# A Model to Account for the Effects of Load Ratio and Hydrogen Pressure on the Fatigue Crack Growth Behavior of Pressure Vessel Steels

**DOI:** 10.3390/ma17174308

**Published:** 2024-08-30

**Authors:** Ashok Saxena, Kip O. Findley

**Affiliations:** 1WireTough Cylinders, Bristol, VA 24201, USA; 2Department of Mechanical Engineering, University of Arkansas, Fayetteville, AR 72701, USA; 3G.S. Ansell Department of Metallurgical and Materials Engineering, Colorado School of Mines, Golden, CO 80401, USA; kfindley@mines.edu

**Keywords:** fatigue crack growth, hydrogen embrittlement, ferritic steels, pressure vessel steels, load ratio

## Abstract

A phenomenological model for estimating the effects of load ratio *R* and hydrogen pressure $P_{H_{2}}$ on the hydrogen-assisted fatigue crack growth rate (HA–FCGR) behavior in the transient and steady-state regimes of pressure vessel steels is described. The “transient regime” is identified with crack growth within a severely embrittled zone of intense plasticity at the crack tip. The “steady-state” behavior is associated with the crack growing into a region of comparatively lower hydrogen concentration located further away from the crack tip. The model treats the effects of *R* and $P_{H_{2}}$ as being functionally separable. In the transient regime, the effects of the hydrogen pressure on the HA–FCGR behavior were negligible but were significant in the steady-state regime. The hydrogen concentration in the steady-state region is modeled as being dependent on the kinetics of lattice diffusion, which is sensitive to pressure. Experimental HA–FCGR data from the literature were used to validate the model. The new model was shown to be valid over a wide range of conditions that ranged between −1≤R≤0.8 and $0.02\leq P_{H_{2}}\leq 103$ MPa for pressure vessel steels.

## 1. Introduction and Background

The cyclic life of pressure vessels for hydrogen storage depends on the maximum service pressure and the amplitude of its fluctuations during service. These vessels are often autofrettaged, which is a process that results in significant compressive residual stresses in the wall of the vessel [1]. For engineering design and remaining life calculations, a model is needed for predicting the hydrogen-assisted fatigue crack growth rate (HA–FCGR) behavior for load ratios (*R*) in the range of −1≤R≤0.8, and hydrogen pressures $P_{H_{2}}$ that range from low values, which do not significantly affect the crack growth behavior, to approximately 100 MPa, where the effects of pressure are quite significant. The ASME Section VIII-Div 3 equation [2,3,4] for modeling the effects of positive *R* values at constant hydrogen pressure and a previous model [5,6] with its recent modification [7] for accounting for the effects of pressure re used as the starting points in this study. Neither of the previous models extended over the entire range of load ratios and hydrogen pressures experienced by the vessels during service.

The model focuses on the crack growth rate *da*/*dN* ranging between 10^−5^ to 10^−2^ mm/cycle, where the HA–FCGR exhibits a two-region behavior [3,4,5,6,7,8,9]. In air or benign environments, this region lies in the regime of the linear relationship between *log*(*da*/*dN*), which is the fatigue crack growth rate per cycle, and *log*(∆*K*), which is the cyclic stress intensity parameter.

Figure 1 shows a schematic comparison of the fatigue crack growth (FCG) behavior in hydrogen and benign environments. In the hydrogen environment, the so-called transient region peels off from the baseline fatigue crack growth rate in air and is characterized by a high slope of the *log*(*da*/*dN*) versus *log*(∆*K*) relationship. It then connects with the steady-state region, where the crack growth rate *da*/*dN* = *X_tr_*, where *X_tr_* is the crack growth rate at which the transient region ends and the steady-state region begins. Such trends have been widely reported in the literature [2,3,4,5,6,7] and are known to have the characteristics described next.

The baseline trend is characterized by the behavior in air, or at very low hydrogen partial pressures that are less than a threshold value. The fatigue crack growth rate *da*/*dN* in this region is given by the well-known Paris law (Equation (1)):(1)$$\frac{da}{dN}=C_{20}^{0}{\left(∆K\right)}^{m_{0}}$$
where $C_{20}^{0}$ and $m_{0}$ are regression constants derived from data of the experimental fatigue crack growth rate in air, as shown in Figure 1. For pressure vessel steels, neither constant depends on the load ratio or the loading frequency [6,7]. The superscript in $C_{20}^{0}$ refers to hydrogen partial pressure of 0; in the subscript, 2 refers to the steady-state region and 0 to a load ratio of 0. Thus, $C_{20}^{0}$ is the pre-exponent constant at *R* and $P_{H_{2}}$ values of 0 in the steady-state region.

The crack growth rate in the transient region ${\left(\frac{da}{dN}\right)}_{tr}$ is characterized by a high slope m_1_ of the *log*(*da*/*dN*) versus *log*(∆*K*) trend and is given by Equation (2):(2)$${\left(\frac{da}{dN}\right)}_{tr}=C_{1}\left(R,P_{H_{2}},\nu \right){\left(∆K\right)}^{m_{1}}$$
where $C_{1}$ depends on the load ratio *R*, the hydrogen pressure $P_{H_{2}}$, and the loading frequency ν; $m_{1}$ is a constant in this region, which is characteristic of a severely embrittled material. The crack growth rate in the steady-state region ${\left(\frac{da}{dN}\right)}_{ss}$ is characterized by Equation (3):(3)$${\left(\frac{da}{dN}\right)}_{ss}=C_{2}\left(R,P_{H_{2}},\nu \right){\left(∆K\right)}^{m_{2}}$$
where the slope of the *log*(*da*/*dN*) and *log*(∆*K*) relationship is *m*_2_ < *m*_1_, and $C_{2}$ is a constant that depends on the load ratio, the hydrogen pressure, and the loading frequency. It is observed in pressure vessel steels that $m_{2}\approx m_{0}$. The value of $C_{2}$ at a load ratio of 0 and hydrogen pressure of 0 is equal to $C_{20}^{0}$, as in Equation (1). These observations guide the development of the model that quantifies the relationship between the HA–FCGR and variables such as the hydrogen pressure, loading frequency, and load ratio for use in structural integrity analyses.

## 2. Model Development and Evaluation

In this section, the phenomena that underlie the HA–FCGR behavior are first discussed. There are two primary mechanisms by which fatigue cracks grow: (a) the pure fatigue mechanism represented by ${\left(\frac{da}{dN}\right)}_{air}$, which exists even without hydrogen and constitutes the baseline fatigue crack growth rate (FCGR) trend for hydrogen pressures below a threshold level, and (b) the hydrogen-assisted fatigue crack growth ${\left(\frac{da}{dN}\right)}_{HA}$ in hydrogen at pressures above the threshold level [8,9]. In the proposed model, the two mechanisms are assumed to operate independently, and the crack growth rate is given by the dominant of the two mechanisms, in other words, by the higher of the two associated rates, as stated in Equation (4). The goodness of this assumption was tested with data and the results are shown in the subsequent sections.
(4)$$\frac{da}{dN}=max\left|{\left(\frac{da}{dN}\right)}_{air},{\left(\frac{da}{dN}\right)}_{HA}\right|$$

Next, we focus on the term ${\left(\frac{da}{dN}\right)}_{HA}$ and its behavior in two regimes, namely, the transient and the steady-state regimes, as seen in Figure 1. The mechanisms that determine the hydrogen content in the steel in the crack tip region and the HA–FCGR regime are schematically shown in Figure 2 and described below.

We assume that the hydrogen-enhanced local plasticity (HELP) and hydrogen-enhanced strain-induced vacancies (HESIVs) [10,11,12,13,14,15] are mechanisms by which hydrogen in high concentrations is quickly swept into the immediate vicinity of the crack tip. This HELP- and HESIV-dominated region shown in Figure 2 and Figure 3 is enabled by an avalanche of new dislocations and vacancy formations in response to the monotonic and cyclic plastic deformation near the crack tip. The dislocation mobility due to the slip-and-vacancy-enabled climb in the zone of intense plasticity instantly carries the hydrogen forward within this region. This region extends through a distance *X_tr_*, which is estimated as being on the order of the crack tip opening displacement (CTOD) given by Equation (5) [5,7]. In this region, there is a third mechanism, namely, hydrogen enhanced decohesion (HEDE), which also operates and reduces the cohesive strength of the metallic bonds causing embrittlement.
(5)$$X_{tr}\sim CTOD\approx \frac{K^{2}}{E\sigma_{0}}$$

The hydrogen concentration in this severely embrittled region is high and primarily consists of trapped hydrogen, i.e., *C_T_*, at dislocations and vacancies, and comparatively small amounts due to lattice diffusion, i.e., *C_L_*; it follows that *C_T_ >> C_L_*. Crack growth rates are expected to be highly sensitive to increases in ∆*K* in this region, as borne out by the experimental observations. The effects of hydrogen pressure on the HA–FCGR in this region is expected to be small for the following reasons: A threshold pressure is required to keep the crack tip continuously supplied with hydrogen, but the transport of hydrogen once it enters the crack tip region depends primarily on the dislocation mobility due to the slip-and-vacancy-enhanced dislocation climb. Thus, increasing the hydrogen pressure beyond the threshold level is not likely to further affect hydrogen concentrations in the zone of intense deformation, and thus, the high hydrogen concentration is limited to the extent of this zone or *X_tr_*.

If *da*/*dN* > *X_tr_*, the crack grows into the material where the hydrogen concentration is assumed to be limited by lattice diffusion, i.e., *C_L_*, as shown in Figure 3, and the increment in *da*/*dN* with ∆*K* is also expected to decrease, as shown schematically in Figure 1. This region is termed as the steady-state region. As stated before, the slopes in the *log* (*da*/*dN*) versus *log* (∆*K*) in the transient and steady-state regions are represented by $m_{1}$ and $m_{2}$, respectively. The pre-exponent constants are *C*_1_ and *C*_2_, as seen in Equations (2) and (3), respectively, and they depend on the load ratio, hydrogen pressure, and loading frequency.

The overall *da*/*dN* is represented by Equation (6) below. The explanation and derivation of Equation (6) are described in Appendix A.
(6)$$\frac{da}{dN}=min\left|{\left(\frac{da}{dN}\right)}_{tr},{\left(\frac{da}{dN}\right)}_{ss}\right|$$

Next, the previously proposed models used to represent the effects of the loading frequency and the load ratio *R* at a constant hydrogen pressure and to represent the effects of hydrogen pressure at constant *R* [5,7] are described. These are then followed by the description of a comprehensive model that concurrently accounts for both the effects of the load ratio and hydrogen pressure on the HA–FCGR behavior.

### 2.1. Pressure Vessel Steels

The model developed in the subsequent sections of this paper applies to ferritic, quenched, and tempered low-alloy steels with yield strengths of 482.5 MPa or higher and ultimate strength levels less than 930 MPa. The microstructure consists of tempered martensite that results from first solution annealing, followed by oil quenching, and then tempering heat treatments. The standard chemical composition and mechanical properties are shown in Table 1 and Table 2, respectively.

### 2.2. The Effects of the Loading Frequency on the HA–FCGR Behavior

It is important to explore the effects of the loading frequency on the HA–FCGR kinetics because it is a consideration when obtaining experimental data relevant to service loading conditions. The experimental results [16,17] suggest that the effects of the loading frequency on the HA–FCGR, if at all present, are small. This was systematically investigated in a previous study [17] by conducting tests at constant ∆*K* at cyclic frequencies ranging from 0.001 to 6 Hz (four orders of magnitude) at *R* values of 0.1 and −1.0, as seen in Figure 4. In the case of *R* = −1.0, ∆*K* = *K_max_* [18]. The results show that any effects of the frequency on the HA–FCGR at both *R* values and multiple ∆*K* values lying in the transient and steady-state regions are not significant. Therefore, the frequency dependence in Equations (2) and (3) is dropped.

The above observation showing the lack of frequency dependence in the HA–FCGR behavior implies that the embrittlement in the near crack tip region caused by hydrogen occurs rapidly. In the transient regime at low ∆*K* levels, where the build-up of hydrogen concentration is due to an increase in dislocation density and mobility, this trend is expected. The crack growth during each cycle in this regime is contained within the severely embrittled zone in which the hydrogen uptake is instantaneous. However, in the steady-state region, where the hydrogen build-up is assumed to be due to lattice diffusion, the lack of frequency dependence is not as obvious. One possibility is that the crack tip stress state in the region beyond *X_tr_* is triaxial and is expected to accelerate the hydrogen diffusion rates, which causes the hydrogen concentration to build rapidly. However, the kinetics of diffusion relative to loading frequency are unknown, and thus, the explanation is speculative and deserves a more in-depth investigation with careful experiments and atomistic modeling. This is left to future studies.

### 2.3. The Effects of Load Ratio R on the HA–FCGR

The effects of the load ratio *R* for 0≤R≤0.8 at a constant hydrogen pressure in the transient and steady-state regions were studied by other researchers [2,3,4,17,19]. The following equations obtained from data generated at a hydrogen pressure of 103 MPa proposed by San Marchi et al. [2,3] are widely used, including in the ASME Design Code [20]. For *R* ≥ 0,
(7)$${\left(\frac{da}{dN}\right)}_{tr}=C_{10}\frac{1+{CH}_{1}R}{1-R}{\left(∆K\right)}^{m_{1}}=C_{10}F(R_{K1}){\left(∆K\right)}^{m_{1}}$$
(8)$${\left(da/dN\right)}_{ss}={C_{20}\frac{1+{CH}_{2}R}{1-R}\left(∆K\right)}^{m_{2}}=C_{20}F(R_{K2}){\left(∆K\right)}^{m_{2}}$$
where $C_{10}$ and $C_{20}$ are the values of *C*_1_ and *C*_2_, respectively, at *R* = 0 and at some specified constant pressure. *CH*_1_ and *CH*_2_ are constants related to the effects of the load ratio in the transient and steady-state regimes, respectively. $F(R_{K1}$) and $F(R_{K2}$) are functions that model the effects of *R* on the HA–FCGR behavior in these regimes. These constants and functions were derived by San Marchi et al. [2] from the regression of data from multiple load ratios for *R* > 0 at a constant hydrogen pressure of 103 MPa and are listed in Table 3. Choosing *da*/*dN* as the lower of the two values predicted from Equations (7) and (8) leads to the behavior plotted in Figure 5 for several load ratios for the hydrogen pressure of 103 MPa. For comparison, the FCGR behavior in the air is also included in Figure 5. The constants for a hydrogen pressure of 10 MPa from Saxena, Nibur, and Prakash [17] are also listed in Table 3. Next, we discuss the HA–FCGR behavior for −1≤R≤0.

The ASME Section VIII-Div 3 code [20] and ASTM Standard E-647 [18] both recommend that for negative load ratios, $∆K=K_{max}$. In other words, the negative portion of the load is not used in calculating the magnitude of ∆*K*. However, the *da*/*dN* relationship with ∆*K* must still be experimentally derived for various negative *R* values. The ASME code [20] currently recommends using the *R* = 0 constants for all *R* < 0. This assumption was experimentally evaluated for a hydrogen pressure of 10 MPa in a previous study [17] and the data were reanalyzed to address this specific question.

To examine the effects of a negative load ratio on the HA–FCGR behavior, we defined a load-ratio-compensated HA–FCGR *da*/*dN** and plotted this as a function of ∆*K*, where *da*/*dN** is defined in the transient and steady-state regimes by Equations (9) and (10), respectively. This allows all available data to be pooled to assess and compare the behavior for *R* < 0 with those at *R* > 0.
(9)$${\left(da/dN\right)}_{tr}^{*}={\frac{1}{F(R_{K1})}\left(\frac{da}{dN}\right)}_{tr}$$
(10)$${\left(da/dN\right)}_{ss}^{*}={\frac{1}{F(R_{K2})}\left(\frac{da}{dN}\right)}_{ss}$$

To implement Equations (9) and (10), the HA–FCGR data at ∆*K* < 13 $MPa\sqrt{m}$ were treated as being in the transient region, and for ∆*K* ≥ 13 $MPa\sqrt{m}$, the data were considered as part of the steady-state region to calculate *da*/*dN*.* The choice of ∆*K* =13 $MPa\sqrt{m}$ was guided by the transition between the transient and steady-state regimes. The results were not very sensitive to the selected value because both trends converged in the transition region. Note also that $F(R_{K1})$ and $F(R_{K2})$ were equal to 1 for R≤0, and thus, *da*/*dN* = *da*/*dN*.* The data at different load ratios at a constant hydrogen pressure can be consolidated into a single trend. Figure 6a shows the HA–FCGR behavior for *R* values of 0.2, 0.1. −0.5, and −1.0 plotted as *da*/*dN* versus ∆*K* and in Figure 6b as *da*/*dN** versus ∆*K* for a pressure vessel steel.

The data in Figure 6 were obtained from single-edge-notch-tension (SEN(T)) specimens designed with rigid ends suitable for tension–compression loading [17]. The *R* = −0.5 and −1.0 data appear to fall on the trend approximated for *R* = 0, as seen in Figure 6a and even more clearly in Figure 6b, where the overall spread in the data was consolidated using the load-ratio-compensated values *da*/*dN** to show the predicted crack growth rate for *R* = 0. There also does not appear to be any measurable differences in the trends between *R* values of -0.5 and −1.0. The various constants for Equations (7) and (8) for a hydrogen pressure of 10 MPa are listed in Table 3. The equation from the regression fit appears to represent all the data for −1≤R≤0.2 quite well. Also, the ASME recommendation [20], which states that for pressure vessel steels, the HA–FCGR behavior of *R* < 0 should be estimated by the data for *R* = 0, was affirmed.

### 2.4. Effects of Hydrogen Pressure on the HA–FCGR Behavior

To model the effects of the hydrogen pressure on the HA–FCGR behavior, we first established whether the effects of the hydrogen pressure and load ratio *R* were functionally separable, as is assumed in Equations (11) and (12):(11)$${\left(\frac{da}{dN}\right)}_{tr}=F\left(R_{K1}\right)f_{tr}\left(P_{H_{2}}\right)C_{10}{\left(∆K\right)}^{m_{1}}$$
(12)$${\left(\frac{da}{dN}\right)}_{ss}=F\left(R_{K2}\right)f_{ss}\left(P_{H_{2}}\right)C_{20}{\left(∆K\right)}^{m_{2}}$$
where $f_{tr}\left(P_{H_{2}}\right)$ and $f_{ss}\left(P_{H_{2}}\right)$ are the functions for the hydrogen pressure dependence for the transient and steady-state regimes, respectively. At constant pressure, the effect of *R* is represented by Equations (7) and (8), which leads to Equations (9) and (10) to obtain *da*/*dN**, which gives the load ratio compensated HA–FCGR behavior in the transient and steady-state regimes, respectively. Furthermore, as mentioned before, *da*/*dN** is essentially the equivalent HA–FCGR behavior at *R* = 0 for which both $F(R_{K1})$ and $F(R_{K2})$ are equal to 1.

In Figure 7, *da*/*dN** is plotted for the data from various load ratios ranging from 0.1 to 0.5 at constant hydrogen pressures of 10, 45, and 103 MPa. It also includes the data in air that represents the crack growth rates that correspond to the threshold level of hydrogen pressure. The *da*/*dN** versus Δ*K* trends for 0.1≤R≤0.5 appeared to collapse into a distinct trend unique to each pressure. This implies that the effects of pressure and load ratio on the crack growth rates were indeed separable, and the model that consisted of Equations (11) and (12) was a good approach. These equations represented the effects of the pressure and load ratio on the HA–FCGR behavior without the need for any interactive terms.

The trends in Figure 7 show that the HA–FCGR behavior did not vary with the hydrogen pressure in the transient region. Thus, $f_{tr}\left(P_{H_{2}}\right)$ in Equation (11) was assigned a value of 1. In the steady-state region, the HA–FCGR behavior did increase systematically with the pressure. The crack growth rates in this region increased while going from the threshold pressure or less to a pressure of 10 MPa, and there was a further increase in *da*/*dN** as the pressure was increased to 45 MPa. The crack growth rates were similar in the steady-state region between the pressures of 103 and 45 MPa, which indicates a tendency for the effects of pressure to saturate. In other words, the steady-state HA–FCGR behavior blended with the data in air for very low pressures and saturated as the pressure approached 45 MPa. This behavior was modeled by writing $f_{ss}\left(P_{H_{2}}\right)$ as in Equation (13), which is a form that naturally has the desired shape:(13)$$f_{ss}\left(P_{H_{2}}\right)=\left(1+\alpha_{2}{\left(ln\frac{P_{H_{2}}}{0.02}\right)}^{n_{2}}\right)$$

With the HA–FCGR trends described above and as seen in Figure 7, $m_{1}\;$and $C_{10}$ were chosen to be independent of the hydrogen pressure. Also, $C_{20}^{0}$ and $m_{2}$ were determined from the FCGR data in air to represent the trend at $P_{H_{2}}=0.02$ MPa, which is the threshold pressure, as suggested by Amaro et al. [5]. Accordingly, $C_{20}$ was given by combining Equations (12) and (13) as follows:(14)$$C_{20}=C_{20}^{0}\left(1+\alpha_{2}{\left(ln\frac{P_{H_{2}}}{0.02}\right)}^{n_{2}}\right)$$

*CH*_1_ and *CH*_2_ in Equations (7) and (8) were assigned the same values across various pressures as part of the assumption that the effects of the load ratio and pressure were separable. As stated earlier, the data at all load ratios for the same pressure were normalized in Figure 7 to the equivalent of *R* = 0 through plots of *da*/*dN** versus Δ*K*. Subsequently, the data were pooled to derive the values of the model constants using the following methodology.

The proposed mathematical functions, Equations (13) and (14), are natural for describing a phenomenon that consists of a threshold hydrogen pressure at which the effects of pressure begin to appear and subsequently approach a saturation level at higher pressures. The threshold value was selected as 0.02 MPa, as suggested by Amaro et al. [5] for ferritic pipeline steels. Further experiments to more rigorously establish the threshold pressure in pressure vessel steels would be useful; however, it is not likely to affect the values of the other constants that were derived from a vast amount of HA–FCGR data at various other pressures. The HA–FCGR behavior at the threshold level of hydrogen pressure was approximated by the behavior in air because that established the asymptotic level that was approached at hydrogen pressures below the threshold. Subsequently, the HA–FCGR rose when the hydrogen pressure was increased beyond the threshold value to 10 MPa and reached saturation at a pressure of 45 MPa, as seen in Figure 7. Accordingly, the full model for representing the HA–FCGR behavior in the transient and steady-state regions as a function of the load ratio and hydrogen pressure is described by Equations (15) and (16), respectively, which follow from the earlier equations:(15)$${\left(\frac{da}{dN}\right)}_{tr}=C_{10}\frac{1+0.43R}{1-R}{\left(∆K\right)}^{m_{1}}$$
(16)$${\left(\frac{da}{dN}\right)}_{ss}={C_{20}^{0}\left(1+\alpha_{2}{\left(ln\frac{P_{H_{2}}}{0.02}\right)}^{n_{2}}\right)\frac{1+2R}{1-R}\left(∆K\right)}^{m_{2}}$$

Consolidation of the data obtained at different load ratios for the same hydrogen pressures into a single trend by plotting *da*/*dN**, as seen in Figure 7, increased the confidence in the analyses. It allowed for a larger pool of data generated at different load ratios for constant hydrogen pressure to be used to determine the regression constants related to the effects of pressure. $C_{20}^{0}$ is the pre-exponent constant in air (or at a $P_{H_{2}}$ of 0.02 MPa or less) that does not depend on the load ratio. Similarly, $C_{10}$ is the pre-exponent constant in the transient region for a load ratio of 0; it was assumed to be the same at various pressures, as explained before. All the data available in the transient region at all hydrogen pressures were pooled to determine the value of $C_{10}$. The value of *m*_1_ was estimated by the regression of these data and was determined to be 6.15, which compared well with the value of 6.5 based on a single pressure of 103 MPa [2].

The average value of *m*_2_ at various pressures, including the value at $P_{H_{2}}=0.02$ MPa, was 3.219, and the value of *C*_20_ obtained from the best fits of the data at individual hydrogen pressures in the steady state was adjusted to account for this common value of *m*_2_. The HA–FCGR data at various pressures were then used to derive the constants $\alpha_{2}$ and $n_{2}$ to complete the estimation of all constants and are shown in Table 4. Figure 7 shows the excellent agreement between the observed and predicted mean values of *da*/*dN** and ∆*K* at various pressures in the transient and steady-state regions for a variety of load ratios and hydrogen pressures.

The predicted HA–FCGR behavior using the constants listed in Table 4 in Equations (13) and (14) at various pressures are plotted in Figure 7, along with the data. The behavior in the transient region is represented by a solid line and the behavior in the steady-state region at various pressures is represented by the dashed lines using the same color scheme as the data at that pressure. The agreements between the experimental trends and the model predictions were excellent. Thus, the model was deemed suitable since it accurately represented the effects of the load ratio and hydrogen pressure on the HA–FCGR behavior. This is further evaluated in the following discussion.

## 3. Discussion of the Model

Figure 8 shows the HA–FCGR compensated for load ratio effects, i.e., *da*/*dN**, in the transient regime using the same data as in Figure 7 with data from different load ratios and pressures identified using distinct symbols. As mentioned before, the HA–FCGR behavior in this region is independent of the hydrogen pressure. It is noted that there was significant scatter in the data as the Δ*K* levels approached the transition point from the transient to the steady-state regions, but no systematic trends within the scatter were observed. Additional studies are needed to explore this further. In pipeline steels, it is noted that the hydrogen pressure does not influence the HA–FCGR behavior in the transient regime in X-52 steels, but it does in X-100 steels [6]. Thus, the observation that the HA–FCGR’s behavior does not vary with pressure in the transient region may be material-specific. There is a need for more studies on pressure vessel steels at various hydrogen pressures in the transient region for more definitive conclusions. This question is explored mechanistically in the following discussion.

A possible explanation for why *C*_10_ does not vary with hydrogen pressure while *C*_20_ does is as follows. If the presence of hydrogen causes embrittlement by reducing the cohesive strength of metallic bonds, it follows that the severity of embrittlement will depend on the concentration of hydrogen, as also stated by various other researchers [5,7,10,11,12,13,14,15]. In the transient region, the hydrogen concentration is primarily associated with the high density of trapped hydrogen due to the HELP and HESIVs, which is enabled by severe plastic deformation. This zone is estimated to be of the size of the crack tip opening displacement (CTOD) [7,21]. In this zone, the HELP and HESIVs are assumed to serve as the mechanisms for the rapid uptake of hydrogen into the crack tip region, and also as mechanisms for transport within the transient region where the dislocation mobility due to slip and climb is high. Past the transient region, hydrogen cannot be transported into the material by the HELP/HESIVs and must be transported by lattice diffusion with kinetics that depend on the hydrogen pressure. Thus, in regions where the HELP/HESIVs dominate, lattice diffusion is expected to only be a secondary contributor to the hydrogen concentration, as shown schematically in Figure 3. It also follows that for HELP/HESIV-induced embrittlement to occur, the pressure must exceed the threshold value to ensure a continuous supply of hydrogen, which is quickly swept into the crack tip region where severe plastic deformation occurs. Increasing the hydrogen pressure beyond the threshold value is not expected to significantly change the hydrogen concentration levels in the transient region. In the transition region between the transient and steady-state behavior, both sources of hydrogen are expected to contribute to the hydrogen concentration, which results in a higher variability in the HA–FCGR behavior, as seen in Figure 8.

In the steady-state region, the hydrogen can only be transported through lattice diffusion. The kinetics of diffusion follow Sievert’s law [22], and therefore, the HA–FCGR is expected to be sensitive to the hydrogen pressure, as schematically shown in Figure 3. The form of the relationship representing the effects of pressure on *C*_20_ was shown in Equation (14). This is different from the power-law form suggested by others [5] and more appropriate considering the trends.

Table 4 also provides constants that represent the 95% confidence interval band by listing the values of the relevant constants representing the upper bound (UB) and lower bound (LB) behaviors. The UB constants are the ones that should be used during the design life/inspection interval calculations.

In Figure 9, the 95% CI upper bound HA–FCGR behavior at various hydrogen pressures from the new model is compared with the ASME Master Curve [2,20] for a pressure of 103 MPa. Both models predict similar HA–FCGR behavior in the steady-state region at 103 MPa, but the new model predictions are below those of the ASME Master Curve in the transient region. Additionally, using the new model will yield design lives of vessels that are higher than those predicted from the ASME recommended constants when the pressures are lower than 103 MPa, which is frequently the case during service. The additional conservatism adds to the manufacturing cost, the weight of hydrogen storage vessels, and maintenance costs during service by requiring more frequent inspections.

## 4. Summary and Conclusions

A model for estimating the effects of the load ratio *R* and hydrogen pressure $P_{H_{2}}$ on the hydrogen-assisted fatigue crack growth rate (HA–FCGR) behavior in the transient and the steady-state regimes of pressure vessel steels is described. The “transient regime” is identified with a region dominated by the hydrogen-enhanced local plasticity (HELP) and hydrogen-enhanced strain-induced vacancies (HESIVs) in which the crack is assumed to grow into a severely embrittled material at the crack tip. The “steady-state” behavior is associated with the crack growing into a region of comparatively lower hydrogen concentration located further away from the crack tip. The newly proposed model accounts for these physical phenomena in its formulation. The following conclusions can be drawn from the study:The proposed model does an excellent job of representing the HA–FCGR data on pressure vessel steels obtained over a wide range of load ratios and hydrogen pressures.The model assumes that the effects of hydrogen pressure and load ratio on the HA–FCGR behavior are separable and the data over a wide range, i.e., −1≤R≤0.5 and $0\leq P_{H_{2}}\leq 103$ MPa, supports this assumption.A load-ratio-compensated HA–FCGR *da*/*dN** was defined, which allows for data at different load ratios to be included in obtaining the model constants related to hydrogen pressure, thereby increasing the fidelity of the model.The HA–FCGR behavior at negative load ratios for a hydrogen pressure of 10 MPa was considered from a previous study in formulating the model. The practice of not including the negative loads in the estimation of Δ*K* was supported by the HA–FCGR data at *R* = −1 and −0.5. The approximated HA–FCGR behavior at *R =* 0 using the model was shown to represent the behavior at all negative load ratios.The HA–FCGR behavior in the transient regime did not show systematic variations due to hydrogen pressure. Higher data scatter in the HA–FCGR was observed in the transition region between the transient and steady-state regions.The mean values and the 95% confidence interval values for the model constants were estimated for use in the structural integrity assessments. The predicted crack growth rates from the model can reduce the conservatism of the model constants recommended in the current version of the ASME Code.

## Figures and Tables

**Figure 1 materials-17-04308-f001:**
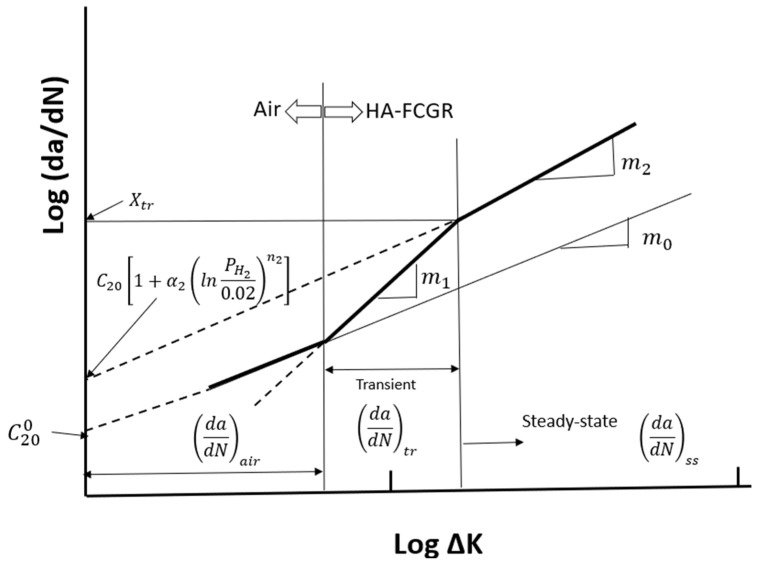
Schematic representation of the two-region hydrogen-assisted fatigue crack growth behavior at a constant pressure and load ratio observed in experimental HA–FCGR data, along with the baseline trend in air.

**Figure 2 materials-17-04308-f002:**
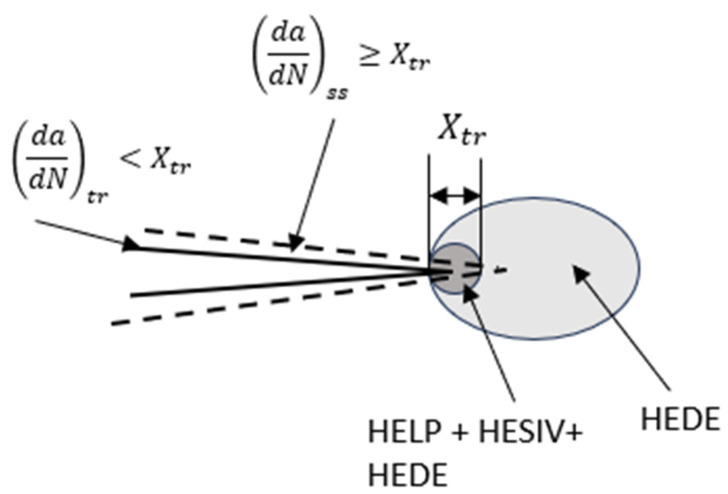
Schematic representation of the severely embrittled crack tip region containing high concentrations of hydrogen enabled by the HELP and HESIVs surrounded by a less embrittled region in which the hydrogen concentration is limited by the hydrogen diffusion kinetics. At lower ∆*K*, in the transient region, the crack extension during one fatigue cycle is less than *X_tr_*, and at higher ∆*K*, the crack extension during the fatigue cycle exceeds *X_tr_*. The diagram is adapted from Amaro et al. [5].

**Figure 3 materials-17-04308-f003:**
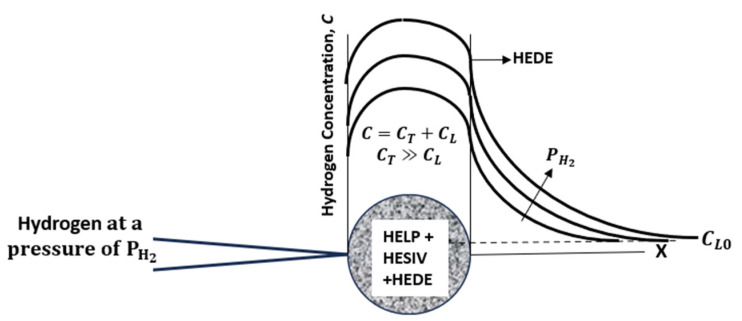
Schematic of the hydrogen concentrations in regions where the HELP and HESIVs determine the hydrogen concentration and in the region beyond which the hydrogen concentration depends on lattice diffusion.

**Figure 4 materials-17-04308-f004:**
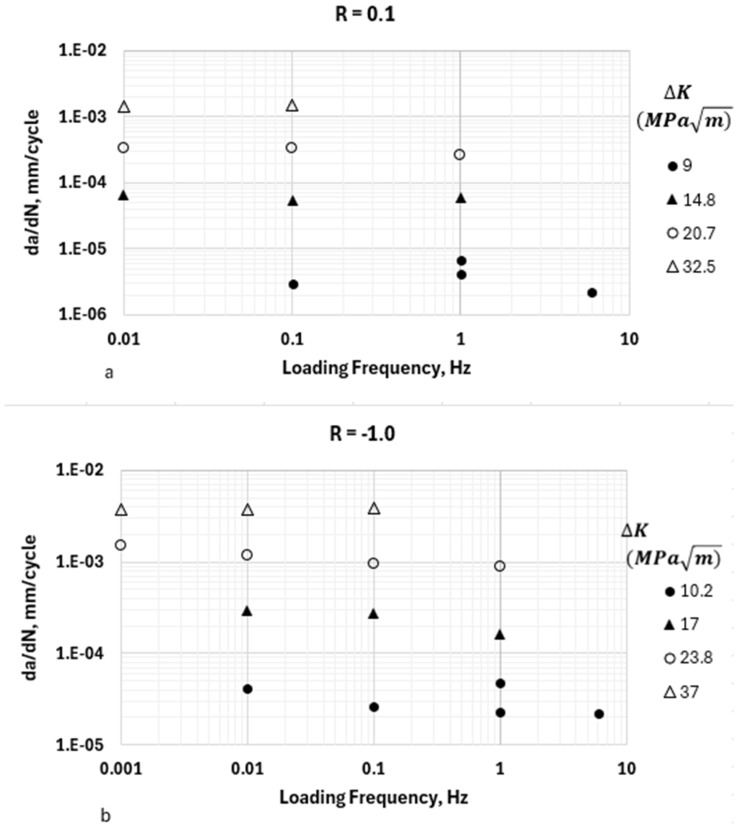
The effects of the loading frequency on the HA–FCGR behavior in pressure vessel steels at (**a**) an *R-*value of 0.1 and (**b**) an *R*-value of −1.0 at a hydrogen pressure of 10 MPa. The data were taken from reference [17].

**Figure 5 materials-17-04308-f005:**
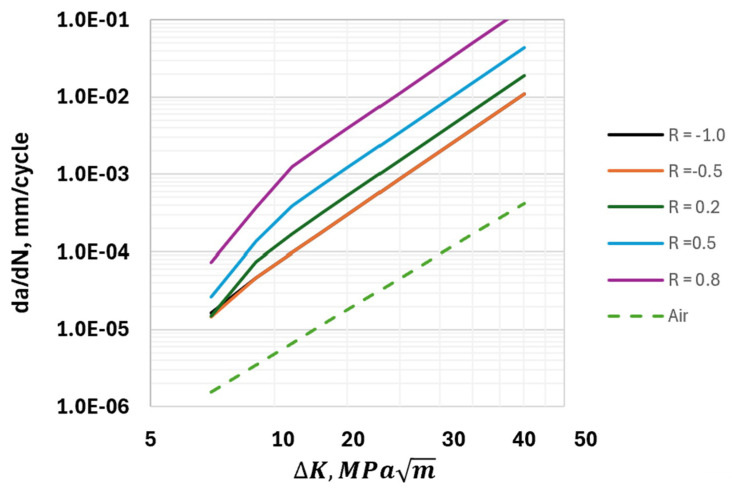
*da*/*dN* versus ∆*K* trends calculated from Equations (8) and (9) using the constants listed in Table 3 for a wide range of *R* values, where −1 ≤ *R* ≤ 0.8, at a hydrogen pressure of 103 MPa; the trend in the air is included for comparison. Also, the trends for *R* = −1.0 and −0.5 are identical.

**Figure 6 materials-17-04308-f006:**
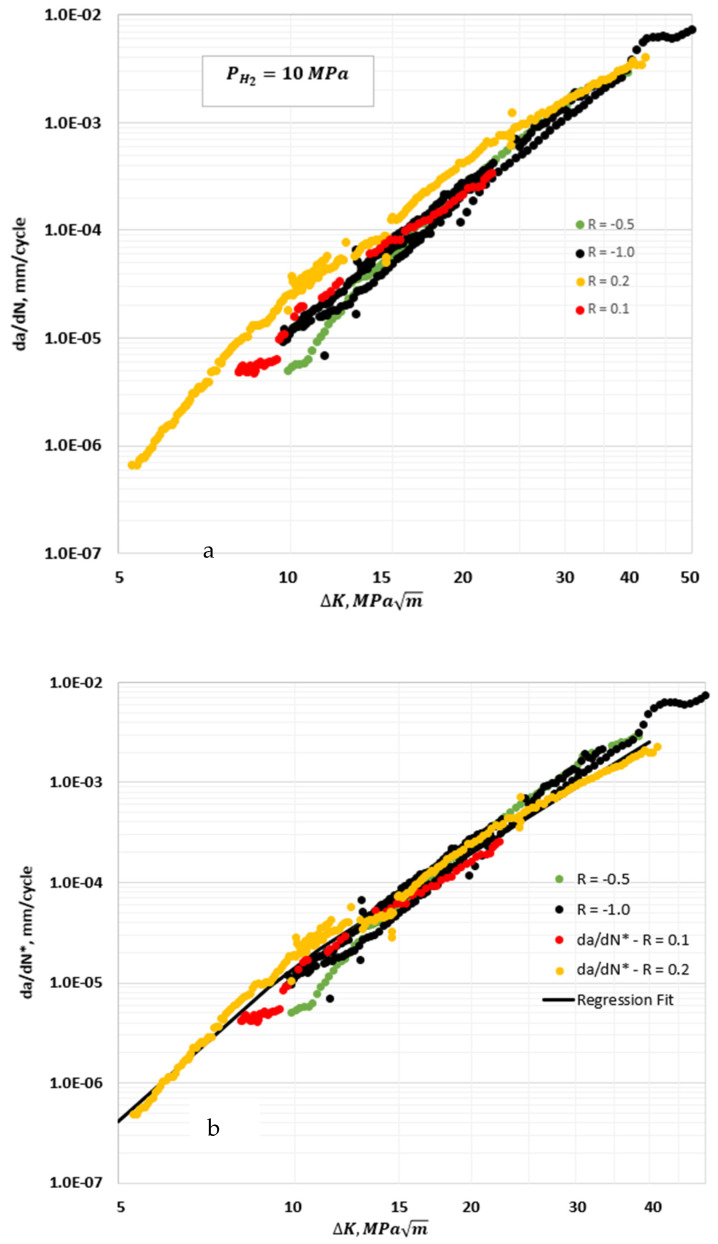
Plots of the HA–FCGR behavior at a hydrogen pressure of 10 MPa for 0.2≤R≤−1.0 (**a**) plotted as *da*/*dN* and (**b**) as load-ratio-compensated rates *da*/*dN**. Data taken from reference [17] were reanalyzed in these figures.

**Figure 7 materials-17-04308-f007:**
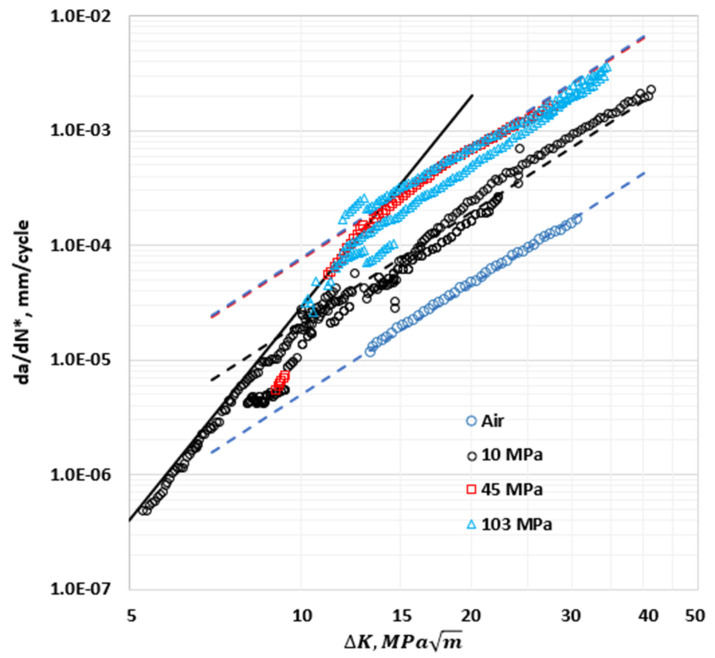
Load-ratio-compensated HA–FCGR behavior of pressure vessel steels at hydrogen pressures of 10, 45, and 103 MPa compared with that in air. All HA–FCGR data used to derive the constants in Equations (14) and (15) are plotted along with the fitted trends at the various pressures. The solid line represents a fit to the data in the transient region at all pressures and the dashed lines using the same color scheme, as the data represent the fitted steady-state behavior for the various pressures using the constants in Table 4.

**Figure 8 materials-17-04308-f008:**
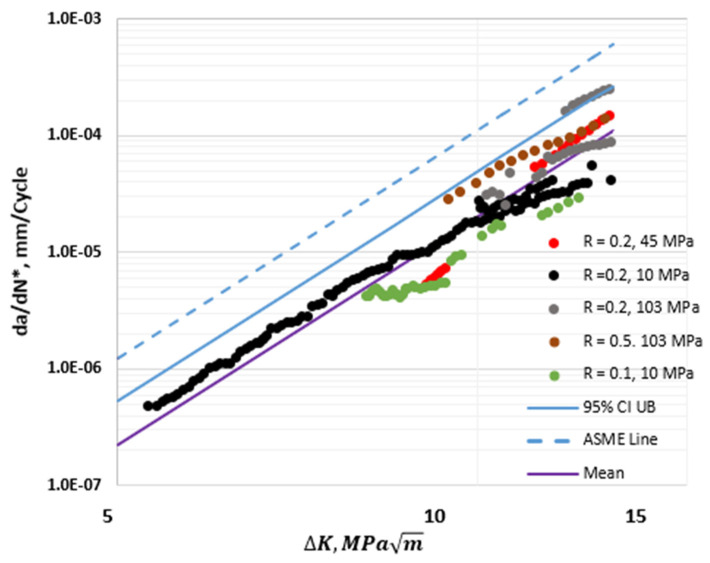
Load-ratio-compensated HA–FCGR behavior at different hydrogen pressures in the transient regime.

**Figure 9 materials-17-04308-f009:**
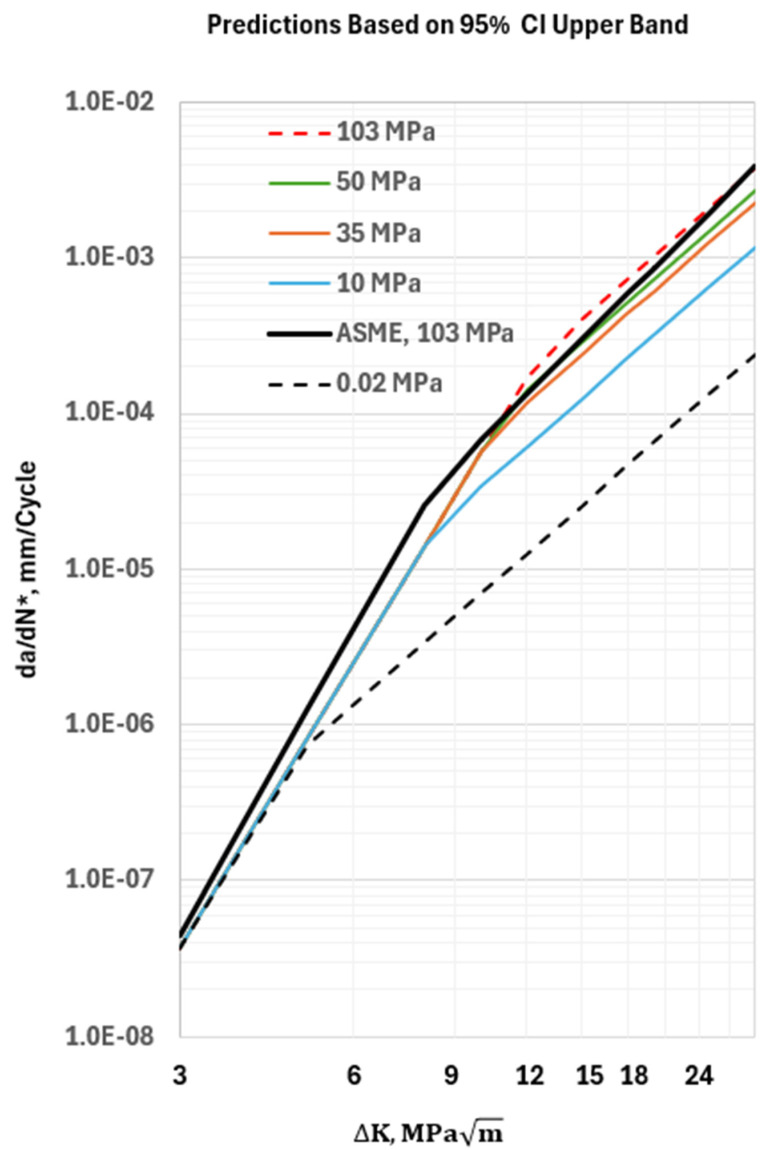
Plots of the 95% CI upper bound of the predicted HA–FCGR behavior from the model at various hydrogen pressures at an equivalent load ratio of 0. The solid black line is the trend from the ASME Section VIII—Division 3 model for a hydrogen pressure of 103 MPa at *R* = 0.

**Table 1 materials-17-04308-t001:** Standard chemical composition (weight %) of the metal liner steel in Type 2 vessels and the vessel itself in Type 1 vessels.

Element	C	Mn	P	S	Ni	Si	Cr	Mo	Cu	Al	Fe
ASME SA-372 Gr J Class 70	0.35–0.50	0.75–1.05	<0.025	<0.025	-	0.15–0.35	0.80–1.15	0.15–0.25	-	-	Bal

**Table 2 materials-17-04308-t002:** Standard tensile and ultimate tensile strengths of ferritic pressure vessel steels used in experimental studies to characterize the HA–FCGR behavior.

Material	0.2% Yield Strength, MPa	Tensile Strength, MPa	% Elongation
SA-372, Grade J Class 70 Standard	482.5 Min	827–930	18 Min

**Table 3 materials-17-04308-t003:** Constants for the HA–FCGR behavior of SA 372 Grade E/J steels at a hydrogen pressure of 103 MPa and 10 MPa. The constants for 0≤R≤0.8 are the same as presented in reference [2], and for −1≤R≤0, the constants were estimated from data available from a previous study [17].

***P*_*H*_2__ = 103 MPa**								
Range of *R*	Transient Regime				Steady-State Regime			
	*C* _10_		*CH* _1_	*m* _1_	*C* _20_		*CH* _2_	*m* _2_
	*da*/*dN* in in/cycle, ∆*K* in $ksi\sqrt{in}$	*da*/*dN* in mm/cycle, ∆*K* in $MPa\sqrt{m}$			*da*/*dN* in in/cycle, ∆*K* in $ksi\sqrt{in}$	*da*/*dN* in mm/cycle, ∆*K* in $MPa\sqrt{m}$		
0≤R≤0.8	2.54 × 10^−12^	3.5 × 10^−11^	0.43	6.5	8.34 × 10^−10^	1.5 × 10^−8^	2.0	3.66
−1≤R≤0			0				0	
***P*_*H*_2__ = 10 MPa**								
0≤R≤0.5	5.83 × 10^−12^	9 × 10^−11^	0.43	5.23	1.68 × 10^−10^	3 × 10^−9^	2.0	3.7
−1≤R≤0			0				0	

**Table 4 materials-17-04308-t004:** Values of constants in Equations (14) and (15). The *da*/*dN* is in mm/cycle and ∆*K* is in $MPa\sqrt{m}$.

	$C_{10}$	$m_{1}$	*CH* _1_		$C_{20}^{0}$	$m_{2}$	*CH* _2_		$n_{2}$	$\alpha_{2}$
			*R* ≥ 0	*R* ≤ 0			*R* ≥ 0	*R* ≤ 0		
Mean	2.0 × 10^−11^	6.15	0.43	0	2.94 × 10^−9^	3.219	2.0	0	4.2	0.0018
95% CI UB	3.94 × 10^−11^	6.15	0.43	0	4.21 × 10^−9^	3.219	2.0	0	4.2	0.0018
95% CI LB	1.01 × 10^−11^	6.15	0.43	0	2.05 × 10^−9^	3.219	2.0	0	4.2	0.0018

## Data Availability

The data generated and used in this study may be available by contacting the first author of the paper at asaxena@uark.edu.

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
