# Peer review of "A Model to Account for the Effects of Load Ratio and Hydrogen Pressure on the Fatigue Crack Growth Behavior of Pressure Vessel Steels"

_materials, 2024, doi:10.3390/ma17174308_

Round 1

Reviewer 1 Report

Comments and Suggestions for Authors

The article presents a model to account for the effects of load ratio and hydrogen pressure on the fatigue crack growth behavior of pressure vessel steels. The work is of the nature of experimental research, but the authors did not avoid some shortcomings and imprecise descriptions. I think the comments listed below, if taken into account and corrected by the authors, will improve the quality of the work.

Suggested changes

1. Please explain all abbreviations and markings at work, e.g. FCG, and others.

2. It is good practice to place a table containing information on basic mechanical properties of material as the yield strength, tensile strength. This data should be in the work.

3. The work is based on experimental research, so photos of cracks in the tested elements should be shown, which will make the presented results credible.

4. A diagram of the tested object would also be welcome. If only steel was tested, please provide a drawing of the tested specimen.

5. Please add the number of specimens tested for a given load and whether there was repeatability.

6. There is a clear lack of a statistical analysis since it seems that no repetitive measurements were performed.

7. How was DK calculated for stress ratio R = -1?

8. How many specimens (tanks) were tested and was a crack observed in each? Were the cracks in the same places?

9. Were brittle, plastic or mixed cracks observed more often?

10. It would be worthwhile to quote also paper of: 1) Experimental and numerical investigation of mixed mode I+II and I+III fatigue crack growth in S355J0 steel. Int. J. of Fatigue, Vol. 113, 2018, pp. 160-170.

Author Response

Reviewer 1 -Comments and response

The article presents a model to account for the effects of load ratio and hydrogen pressure on the fatigue crack growth behavior of pressure vessel steels. The work is of the nature of experimental research, but the authors did not avoid some shortcomings and imprecise descriptions. I think the comments listed below, if taken into account and corrected by the authors, will improve the quality of the work.

Suggested changes

  1. Please explain all abbreviations and markings at work, e.g. FCG, and others.

All abbreviations have been explained, the first time that they appear in the manuscript. We had missed a couple in the original version which have now been fixed.

  1. It is good practice to place a table containing information on basic mechanical properties of material as the yield strength, tensile strength. This data should be in the work.

No new experiments were performed as part of this study; all data were taken from the prior studies and references to those studies have been made where the details requested are spelled out. Nevertheless, we have added two tables and the accompanying descriptions that provide the generic material chemistry and properties of pressure vessel steels.

  1. The work is based on experimental research, so photos of cracks in the tested elements should be shown, which will make the presented results credible.

Again, the work was based on data published in the literature. The model is new, but the data used to validate the model is from the literature that has been cited. No new experimental data was generated as part of this study.

  1. Please add the number of specimens tested for a given load and whether there was repeatability.

See the answer to comment 3 above.

  1. There is a clear lack of statistical analysis since it seems that no repetitive measurements were performed.

I am not sure what the reviewer means by this comment. The data sets used contained data from multiple specimens as part of prior work by one of the current authors but also by Sandia National Laboratory which made the data available to the authors. All the data has been previously published and references are provided. Statistical analyses of the results are included in the results in the form of confidence intervals. Also, in all cases, several specimens were tested under identical conditions, so the results are statistically significant.

  1. How was DK calculated for stress ratio R = -1?

As clearly explained in lines 226 and 22 of the original manuscript, ΔK for R of -1 and -0.5 is the Kmax value as per ASTM standard.

  1. How many specimens (tanks) were tested and was a crack observed in each? Were the cracks in the same places?

Cracks were in the test specimens and not in tanks. In test specimens, the cracks are always machined in the same location.

  1. Were brittle, plastic or mixed cracks observed more often?

The crack growth occurred via a hydrogen-assisted fatigue crack growth mechanism and not by fracture. Subcritical fatigue crack growth is not classified as brittle, ductile, or mixed growth, so the comment is not clearly stated.

  1. It would be worthwhile to quote also paper of: 1) Experimental and numerical investigation of mixed mode I+II and I+III fatigue crack growth in S355J0 steel. Int. J. of Fatigue, Vol. 113, 2018, pp. 160-170.

We checked the reference that the reviewer wants us to quote. The reference is about mixed-mode fatigue crack growth that has no relevance to the work presented in this paper and is therefore not included.

Reviewer 2 Report

Comments and Suggestions for Authors

The manuscript proposes a model for estimating the effects of load ratio and hydrogen pressure on the hydrogen-assisted fatigue crack growth rate behavior in the transient and steady-state regimes of pressure vessel steels. The paper is well-written, and its structure is well-organized. It is recommended that it be accepted after minor revisions.

1. There is a formatting issue in line 107. Please revise it.

2. There is a formatting issue in line 170. Please revise it.

3. There is a formatting issue in line 425. Please revise it.

4. Please adjust the format of the table in lines 241-242; the current presentation is not clear enough.

5. It is suggested that the format of the two graphs on the left and right in Figure 2 be unified, including the position of the letters and the annotation of Xtr.

6. In the first paragraph of Section 2, could you please provide references to support the assumption that the two mechanisms are independent of each other?

7. The comparison between the model's predicted results and the experimental data, as well as the degree of error, is not clear enough. It is suggested that the figures and their captions be revised.

Author Response

Reviewer 2 Comments and Responses

The manuscript proposes a model for estimating the effects of load ratio and hydrogen pressure on the hydrogen-assisted fatigue crack growth rate behavior in the transient and steady-state regimes of pressure vessel steels. The paper is well-written, and its structure is well-organized. It is recommended that it be accepted after minor revisions.

  1. There is a formatting issue in line 107. Please revise it.

Thank you, done.

  1. There is a formatting issue in line 170. Please revise it.

Thank you, done

  1. There is a formatting issue in line 425. Please revise it.

Thank you, done

  1. Please adjust the format of the table in lines 241-242; the current presentation is not clear enough.

Thank you, done

  1. It is suggested that the format of the two graphs on the left and right in Figure 2 be unified, including the position of the letters and the annotation of Xtr.

Thank you, done

  1. In the first paragraph of Section 2, could you please provide references to support the assumption that the two mechanisms are independent of each other?

This is an assumption within the proposed model.  We have made that clearer by adding language in the narrative. The goodness of the assumption is embedded in whether the model is successful in representing the HA-FCGR behavior.  Since the model is very successful in accurately representing the experimental trends, the assumptions are supported by the data. We have made changes to the manuscript in a couple of places to respond to the comment.

  1. The comparison between the model's predicted results and the experimental data, as well as the degree of error, is not clear enough. It is suggested that the figures and their captions be revised.

We understand and agree with the reviewer’s comment. Accordingly, the figure caption has been changed substantially and the description of Fig. 7 has been extensively in response to the comment.

Round 2

Reviewer 1 Report

Comments and Suggestions for Authors

The authors took into account the Reviewer's comments and suggestions. I recommend publishing the article in the Materials journal.

Author Response

Comment 1: See response  provided.

Comment 2: Done

Comment 3: Done

Comment 4: Done